# PEGylation Prolongs the Half-Life of Equine Anti-SARS-CoV-2 Specific F(ab’)_2_

**DOI:** 10.3390/ijms24043387

**Published:** 2023-02-08

**Authors:** Mengyuan Xu, Jinhao Bi, Bo Liang, Xinyue Wang, Ruo Mo, Na Feng, Feihu Yan, Tiecheng Wang, Songtao Yang, Yongkun Zhao, Xianzhu Xia

**Affiliations:** 1College of Veterinary Medicine, Jilin University, Changchun 130062, China; 2Changchun Veterinary Research Institute, Chinese Academy of Agricultural Sciences, Changchun 130012, China; 3Center of Infectious Disease Research, Westlake University, Hangzhou 310024, China; 4College of Wildlife and Protected Area, Northeast Forestry University, Harbin 150040, China; 5College of Animal Science and Technology, Jilin Agricultural University, Changchun 130118, China

**Keywords:** SARS-CoV-2, equine polyclonal antibody, specific F(ab’)_2_, PEGylation, pharmacokinetic

## Abstract

Therapeutic antibodies-F(ab’)_2_ obtained from hyperimmune equine plasma could treat emerging infectious diseases rapidly because of their high neutralization activity and high output. However, the small-sized F(ab’)_2_ is rapidly eliminated by blood circulation. This study explored PEGylation strategies to maximize the half-life of equine anti-SARS-CoV-2 specific F(ab’)_2_. Equine anti-SARS-CoV-2 specific F(ab’)_2_ were combined with 10 KDa MAL-PEG-MAL in optimum conditions. Specifically, there were two strategies: Fab-PEG and Fab-PEG-Fab, F(ab’)_2_ bind to a PEG or two PEG, respectively. A single ion exchange chromatography step accomplished the purification of the products. Finally, the affinity and neutralizing activity was evaluated by ELISA and pseudovirus neutralization assay, and ELISA detected the pharmacokinetic parameters. The results displayed that equine anti-SARS-CoV-2 specific F(ab’)_2_ has high specificity. Furthermore, PEGylation F(ab’)_2_-Fab-PEG-Fab had a longer half-life than specific F(ab’)_2_. The serum half-life of Fab-PEG-Fab, Fab-PEG, and specific F(ab’)_2_ were 71.41 h, 26.73 h, and 38.32 h, respectively. The half-life of Fab-PEG-Fab was approximately two times as long as the specific F(ab’)_2_. Thus far, PEGylated F(ab’)_2_ has been prepared with high safety, high specificity, and a longer half-life, which could be used as a potential treatment for COVID-19.

## 1. Introduction

Coronavirus Disease 2019 (COVID-19) is an acute infectious disease caused by SARS-CoV-2. The early and most common manifestations are dry cough, fever, fatigue, headaches, and myalgia. Since the epidemic of COVID-19, the widespread pandemic of it and rising mortality rates have become significant public health concerns [1,2,3]. Therefore, effective, specific, and quickly accessible drugs are needed to control this situation.

Several vaccines have been used to prevent the epidemic of SARS-CoV-2, but they are not completely effective. It could be stated that therapeutic agents will continue at the center of the fight against COVID-19 [4,5,6]. Antibody drugs are effective in treating emerging infectious diseases [7]. Monoclonal antibody drugs have the advantages of high stability and specificity. However, they also have the disadvantages of a long production cycle, high production cost, and easy loss of antigen epitopes [8]. Polyclonal antibody serum product developed in recent years shows good application prospects in various studies. Convalescent plasma is one of the most extensive therapies researched for SARS-CoV-2 and is more likely to produce specific antibodies [9]. Even so, the ability to obtain plasma depends on voluntary donations from recovered patients. In addition, there are significant differences in the antibodies produced by individuals, which may lead to poor treatment effectiveness. These disadvantages could be overcome by using heterologous antibodies obtained from horse plasma immunized with SARS-CoV-2. It has multiple antigen-binding sites, therefore it has a greater ability to neutralize viruses than monoclonal antibodies [10,11]. Equine polyclonal antibody fragments (EpAbs)-F(ab’)_2_ have been proven to be a safe immunotherapy with decades of production experience, which make EpAbs a potential emergency aid to help fight COVID-19 [12,13]. Israel and other countries are stepping up the development of equine antiserum. In addition, Argentina has used it as an emergency-approved drug for clinical treatment [14]. Nevertheless, direct injections will cause an allergic reaction caused by Fc fragments from IgG. F(ab’)_2_ not only retains the activity of IgG but also reduces antibody-dependent cellular cytotoxicity (ADCC) and complement-dependent cytotoxicity (CDC). Therefore, preparation of specific F(ab’)_2_ fragments lacking Fc fragments would be a desired candidate for therapy [15,16]. Compared with a whole IgG, the F(ab’)_2_ possesses the advantage of a small size, enabling better transportation or penetration in vivo to the target, leading to increased therapeutic effects [17]. Nevertheless, one of the concerns related to F(ab’)_2_ is that the reduced hydrodynamic size results in a short circulation half-life due to fast clearance through the kidneys [18].

Polyethylene glycol (PEG) is a nontoxic, harmless chemical that the Food and Drug Administration (FDA) has approved. PEG is applied to foods, cosmetics, and pharmaceuticals. PEGylation is the process of coupling PEG with proteins in order to protect the proteins from hydrolytic degradation, reducing non-specific interactions, increasing solubility, and, most importantly, enhancing their half-life. About 20 PEGylated drugs have been approved by FDA all over the world, and nearly a thousand of them are under development [19,20]. With the development of PEGylated drugs in recent years, they have a broad market. Various PEGylated drugs are shown as follows (Table 1).

PEGylation can be performed randomly (nonselective) or site-directed. However, for F(ab’)_2_, random modification greatly reduces or even inactivates the activity [21,22]. Cysteines (Cys) in protein are generally suitable targets for site-directed labeling with maleimide-activated PEG. PEGylation can be used to target the thiol group of Cys in the hinge region, which is a good idea for site-directed modifications. Generally, thiol groups are obtained by reducing the S–S bond, or they could be genetically engineered to be introduced into antibodies at specific sites. When the S–S bond is reduced, the natural S–S bond is replaced by PEG-linkage [23,24]. The most obvious advantage of this strategy in comparison to traditional PEG conjugation is the chemical linkage that is formed between both chains contributing to the stabilization of the molecular structure [25]. In addition, different molecular weights (MW) or PEG shapes would also significantly impact the half-life and activity. It is well known that the in vitro activity of PEGylated protein decreases with the increase of PEG MW. For example, Stephen Brocchini modified mmTRAILA with 5 KDa, 10 KDa, and 20 KDa PEG, and their half-lives were extended from less than 10 min to 30 min, 350 min, and 400 min, respectively [26]. However, the data displayed that the MW of PEG exceeding 60 KDa could introduce a question of safety because it could not be eliminated [27]. Briefly, the MW of PEG and the site where the PEG binds to the protein are significant.

Based on the background illustrated above, the total IgG in the serum was separated by affinity chromatography and then digested with pepsin to obtain the total F(ab’)_2_. The specific F(ab’)_2_ was obtained using an immunochromatographic column bound to the SARS-CoV-2 RBD. SDS-PAGE indicated that the MW was accurate, and thin layer scanning showed a high purity. In addition, ELISA and pseudovirus neutralization assay showed that the specific F(ab’)_2_ had higher neutralizing activity and affinity than the total F(ab’)_2_. We have evaluated the effects of PEGylation on the activity and half-life of the specific F(ab’)_2_ obtained. The specific F(ab’)_2_ was restored to Fab’ by TCEP and bound to 10 KDa MAL-PEG-MAL immediately (Figure 1). The MW, activity, and pharmacokinetic parameters of the PEGylated F(ab’)_2_ were investigated in detail. We discovered that the MW and half-life of the PEGylated F(ab’)_2_ increased compared to the specific F(ab’)_2_, although the activity had reduced slightly. Finally, the equine anti-SARS-CoV-2 specific F(ab’)_2_ was prepared successfully, and it was safe, efficient, and had a longer half-life after PEGylation.

## 2. Result

### 2.1. Expression and Purification of SARS-CoV-2 RBD

To obtain the specific F(ab’)_2_, SARS-CoV-2 RBD was selected as the antigen to prepare an immunoaffinity chromatography column. The pET-30a(+)-RBD was successfully constructed by connecting the correct RBD gene sequence with DH5α, which was verified by enzyme digestion (Appendix A). After induced expression and the purification of pET-30a(+)-RBD plasmid in *Escherichia coli*, the RBD protein was finally prepared. Then the result of the SDS-PAGE (Figure 2a) and WB (Figure 2b) displayed the MW of the SARS-CoV-2 RBD was correct, as a single band appeared at 32 KDa and had an excellent specificity. The thin layer scanning result showed the purity of the RBD was 94.5% (Figure 2c). Finally, the immunoaffinity chromatography column with RBD as antigen was successfully prepared at pH = 9, 4 °C for 18 h.

### 2.2. Preparation of Specific F(ab’)_2_

The total IgG was successfully purified by the Protein G column, and the MW was displayed at 250 KDa (Figure 3a, lane 1), and then digested with pepsin to obtain the total F(ab’)_2_. The optimum digestion condition was 30 °C for 4 h (data not provided). Finally, the digestion solution was purified by the Protein A column and there was a single target band at 110 KDa (Figure 3a, lane 2). At the same time, the specific F(ab’)_2_ was obtained by immunoaffinity chromatography and was correctly identified by non-reduced SDS-PAGE (Figure 3b). There was a high purity of 96.771% by thin layer scanning (Figure 3c), however, the yield was 4.89% (Table 1).

### 2.3. Comparison of the Affinity and Activity for Specific F(ab’)_2_ and Total F(ab’)_2_

The ELISA(Figure 4a) results show that the concentration was proportional to the absorbance at OD_450_, and that the absorbance of the specific F(ab’)_2_ was higher than that of the total F(ab’)_2_ at the same concentration. In particular, the pseudovirus neutralization assays displayed a similar trend to ELISA. Half maximal inhibitory concentration (IC_50_) of the specific F(ab’)_2_ and the total F(ab’)_2_ were 0.862 μg/mL and 3.492 μg/mL, respectively (Figure 4b). All these data indicate that the specific F(ab’)_2_ had higher affinity and activity to SARS-CoV-2 RBD than the total F(ab’)_2_.

### 2.4. The Recovery of Specific F(ab’)_2_ and Total F(ab’)_2_

The concentrations were detected by BCA, and were 8.58 mg/mL, 6.77 mg/mL, and 1.05 mg/mL of the total IgG, the total F(ab’)_2_, and the specific F(ab’)_2_, respectively. The yield of the total IgG was 100% of the standard, hence the yield of the total F(ab’)_2_ and the specific F(ab’)_2_ was 78.9% and 4.89%, respectively (Table 2).

### 2.5. Preparation and Purification of PEGylated F(ab’)_2_

In the process of reducing the specific F(ab’)_2_ by TCEP, the Fab’ (~50 KDa) was obtained (Figure 5b, lane1). At the same time, 10KDa MAL-PEG-MAL was attached to the reduced Fab’ and there were two conjugates. The Fab-PEG-Fab (Figure 5b, lane 2) and the Fab-PEG (Figure 5c, lane 1) were purified to homogeneity by anion exchange chromatography. During the experiment, the mole ratio of MAL-PEG-MAL:specific F(ab’)_2_ was 4:1, 2:1, and 1:1, respectively (Figure 5a), and the optimum reaction conditions were the mole ratio of MAL-PEG-MAL:specific F(ab’)_2_ = 4:1.

### 2.6. Compared the Affinity and Activity of Specific F(ab’)_2_, PEGylated F(ab’)_2_, and Total F(ab’)_2_

In order to explore the effect of PEGylation on the activity of the specific F(ab’)_2_ in vitro, the affinity ability with the SARS-CoV-2 RBD of the specific F(ab’)_2_, the Fab-PEG-Fab, the Fab-PEG, and the total F(ab’)_2_ were evaluated by ELISA. The increased concentration caused the increase in absorbance at OD_450_. The crave of the specific F(ab’)_2_ was higher than the Fab-PEG and the Fab-PEG-Fab at the same concentration, and the crave of the total F(ab’)_2_ was lower than the Fab-PEG-Fab and higher than the Fab-PEG (Figure 6a). Meanwhile, the pseudovirus neutralization assay showed the result was similar to ELISA. In particular, the IC_50_ of the Fab-PEG-Fab was 5.52 μg/mL, which was lower than the specific F(ab’)_2_ (2.62 μg/mL) and was higher than the total F(ab’)_2_ (9.17 μg/mL) and the PEG-Fab (17.69 μg/mL) (Figure 6b). The results indicate that there was an effect to the affinity and activity when the specific F(ab’)_2_ combined with the PEG. This might be due to the fact that the PEG blocked the binding site of the specific F(ab’)_2_ to the antigen.

### 2.7. Pharmacokinetic Properties

In order to obtain the pharmacokinetic parameters, the specific F(ab’)_2_, Fab-PEG-Fab, and PEG-Fab were administered to SD rats at 1 mg/kg through intraperitoneal injection. The serum was separated and the pharmacokinetic parameters of the specific F(ab’)_2_, Fab-PEG-Fab, and Fab-PEG were determined by ELISA at the same conditions after the blood was collected at different time points (Figure 7a). The results showed that, compared to the specific F(ab’)_2_, the Fab-PEG-Fab displayed a significantly increased half-life, which indeed went from 38.32 h (specific F(ab’)_2_) to about 71.41 h (Fab-PEG-Fab) and the half-life of Fab-PEG was 26.73 h (Figure 7b). Other pharmacokinetic parameters were also obtained (Table 3). In the Summary, there are two products after PEGylation in which the Fab-PEG-Fab had a 1.86-fold increase in half-life and a 1.73-fold increase in the area under the curve (AUC).

## 3. Discussion

Antibodies, convalescent human plasma, and animal-derived high immunoglobulins have generally been several specific antiviral therapies available, in addition to vaccination, to control the epidemic of COVID-19 [28]. Although these methods are effective in prevention and treatment, COVID-19 is still spreading globally, constituting a major public health concern. The urgency to make life-saving treatments for COVID-19 available precludes traditional drug discovery paths, given their typically protracted timelines. Animal-derived immunoglobulin passive therapy is a better option for low-income countries, with the advantages of low cost, fast results and good efficacy. Immunotherapies with hyperimmune serum have been applied as a therapeutic approach in the treatment of coronavirus since 2005. In particular, antibody drugs with F(ab’)_2_ as the main component have been used to treat snake venom, scorpion venom, and tetanus, and achieved tremendous curative effects [18,29,30]. However, the heterogeneous material injected frequently can still bring different effects to the body and even death. Moreover, compared with IgG, F(ab’)_2_ has a shorter elimination half-life in vivo, which leads to both mental and material pressure in patients. Therefore, it was urgent to prepare equine anti-SARS-CoV-2 specific F(ab’)_2_ with high specificity, high safety, and extended serum half-life.

Considering the enormous potential of antibody-based therapy, we developed SARS-CoV-2 specific immunoglobulin fragments F(ab’)_2_. In the present study, on one hand, specific F(ab’)_2_ was obtained from equine hyperimmune serum by self-made immunoaffinity chromatography. On the other hand, the half-life of specific F(ab’)_2_ was extended by site-directed coupling 10 KDa MAL-PEG-MAL. For specificity, the pseudovirus neutralization assay showed that the inhibition rate of specific F(ab’)_2_ against the pseudovirus was significantly higher than that of the total F(ab’)_2_, which was consistent with the trend of ELISA. On the serum circulation time, the half-life of Fab-PEG-Fab was 71.41 h and the half-life of specific F(ab’)_2_ was 38.32 h, which extended 1.86 times. The final experimental data also verified that our results were in line with the expectations, which also suggests that the solution is feasible and credible.

To obtain the equine anti-SARS-CoV-2 specific F(ab’)_2_, we used prokaryotic expression of SARS-CoV-2 RBD as an antigen. According to the final experimental results, the immunoaffinity chromatography column prepared with RBD could successfully prepare specific F(ab’)_2_, which indicates that it is feasible to use RBD as the antigen. The main reason for this phenomenon is that RBD is a crucial site for binding with receptor ACE2, so these F(ab’)_2_ specifically target the RBD and prevent the binding of the virus to its receptor, ACE2. The purity and specificity of specific F(ab’)_2_ obtained with RBD as an antigen was high. However, the yield was low, which might be due to the fact that, although RBD was the key site for the binding of SARS-CoV-2 to the host receptor, it was not a single site that could wholly extract the anti-SARS-CoV-2 specific antibody in horse serum. From this view, it might be better to use full-length S protein as an antigen for immunoaffinity chromatography. However, relevant data also displayed that the specificity of the antibodies extracted from full-length S protein as an antigen was lower than that of RBD, which requires further exploration.

The composition of the PEGylated F(ab’)_2_ was different when different amounts of PEG reacted with the specific F(ab’)_2_. There were two products when the PEG reacted with the reduced F(ab’)_2_. One of the products was Fab-PEG-Fab, in which PEG was used as a scaffold molecule to link two Fab’ together to give Fab-PEG-Fab, and another was a Fab’ combined with a PEG to produce Fab-PEG. This was owing to different amounts of PEG and Fab’ combined to obtain different products. In the process of product separation, due to the shielding of PEG, Fab-PEG-Fab had a higher affinity than PEG-Fab. Finally, Fab-PEG-Fab was eluted after Fab-PEG. In addition, the apparent MW of Fab-PEG-Fab and Fab-PEG were higher than the theoretical MW, because PEGylation would increase the molecular radius of the molecule. However, the ELISA and pseudovirus neutralization assay results confirmed that the affinity and the neutralization activity of the PEGylated F(ab’)_2_ to the antigen were decreased, but the effect was slight. This was due to the high flexibility of the PEG chains swept around the protein, which shielded the interaction capabilities of the protein that were responsible for its in vitro biological function, thus decreasing the in vitro activity [31,32].

In addition, the MW of Fab-PEG-Fab was 140 KDa, and its half-life was 71.41 h. The MW of the specific F(ab’)_2_ was 110 KDa, and its half-life was 38.32 h. This also clearly demonstrates the strong dependence of half-life on molecular size. Remarkably, the molecular size and half-life correlation was almost linear between the specific F(ab’)_2_ and the PEGylated F(ab’)_2_. Nevertheless, the excessive molecular weight would increase the toxic effects on the body. The pharmacokinetic properties might also have relevance in providing a warning for the use of PEGylated F(ab’)_2_, whose pros and cons compared to the full-sized molecules and antibody fragment should be equally considered in a potential development program.

## 4. Materials and Methods

### 4.1. Materials

The IPTG and TMB were purchased from Beijing Solaibao Technology Co., Ltd. (Beijing, China). The Ni-NTA Resin was purchased from Beijing Quanshi Gold Biotechnology Co., Ltd. (Beijing, China). The pepsin was obtained from the Promega company in the United States (Madison, WI, USA). The HiTrap protein a HP and HiTrap protein G HP was obtained from Healthcare in the United States (New York, NY, USA); 30 KDa and 10 KDa ultrafiltration centrifuge tubes were purchased from Merck (Darmstadt, Germany). The TCEP was obtained from Beyotime Biotechnology (Beijing, China). The 10 KDa MAL-PEG-MAL were purchased from Beijing Jiankai Technology Co., Ltd. (Beijing, China). The purification instrument was obtained from Shanghai Jinpeng Analytical Instrument Co., Ltd. (Shanghai, China).

### 4.2. Animals, Cells, and Virus Strains

Sprague Dawley (SD)male rats were purchased from Beijing Weitong Lihua Laboratory Animal Technology Co., Ltd. The SARS-CoV-2 pseudovirus, 293T cells, and 293T-hACE2 cell lines were obtained from the Laboratory of Animal Virology and Special Animal Diseases, Changchun Veterinary Research Institute.

### 4.3. Preparation and Purification of SARS-CoV-2 RBD

The forward primers and reverse primers were designed according to the SARS-CoV-2 RBD gene sequence (amino acids 319–541 in GenBank) and the target fragment was amplified by PCR (polymerase chain reaction). The PCR products were linked with pET30a(+) by T4 ligase and transformed into competent cell (DH5α). Finally, the correct recombinant plasmid was transformed into DH3 and expression was induced by IPTG for 5 h. In addition, the bacterial solution was collected for centrifugation and the precipitation was suspended in an equilibrium solution. Then ultrasonic crushing was performed on ice until the liquid clarified and the supernatant was collected after centrifugation. Finally, the packing was combined with the supernatant at 4 °C overnight and purified by Ni-NTA. The MW was detected by SDS-PAGE and the purity was measured by thin layer scanning.

### 4.4. Preparation of SARS-CoV-2 RBD Immunoaffinity Chromatography Column

The Sepharose 4B gel was rinsed with ddH_2_O, then the gel was activated with 0.2 M Na_2_MnO_4_ for 1 h at room temperature. The activated gel was then balanced by an equilibrium buffer (0.2 M NaHCO_3_ + 0.5 M NaCl, pH = 10). Finally, RBD was combined with gel at 4 °C overnight. The next day, the gel was resuspended with 2% NaBH_4_, and the reaction was turned slowly at room temperature for 1 h, rinsed with a buffer, and drained of excess liquid. The gel was resuspended with 0.1 M Tris-HCl (pH = 8.0) and turned for 2 h at room temperature. Finally, the column bed was balanced by 0.02 M PBS (pH = 7.4), and finally stored at 4 °C with 20% ethanol.

### 4.5. Purification of Total IgG

Equine anti-SARS-CoV-2 serum was diluted 10 times with 0.02 M PBS (pH = 7.2). After the column bed was balanced with 0.02 M PBS, the serum diluent was repeatedly sent through the Protein G column 2–3 times and the fluxion liquid was collected. A total of 0.02 M PBS was used to balance the column bed again, and 0.1 M Glycine-HCL (pH = 2.7) was used to elute the bed. A total of 1 M Tris-Hcl (pH = 9) was added instantly after the eluent was collected until the pH = 7.0. After dialysis at 4 °C overnight, the concentration was calculated by BCA, the MW was calculated by 8% non-reducing SDS-PAGE, and the purity was calculated by thin layer scanning. Finally, the prepared IgG was stored at −20 °C.

### 4.6. Preparation of the Total F(ab’)_2_

To obtain the total F(ab′)_2_, pepsin was dissolved in 0.02 M PBS (pH = 3.2) and was added to the total IgG to digest for 4 h at 30 °C. During this time, samples were taken every hour to determine the optimal reaction time. Finally, the reaction was terminated with NaOH to adjust the pH = 7.0.

### 4.7. Purification of Total F(ab’)_2_

The digestion solution described above was filtered by a 0.22 μm filter membrane and purified by a Protein A column. Briefly, the Protein A column was balanced with 0.02 M PBS (pH = 7.4), then the digestion solution was repeatedly sent through the Protein A column 2–3 times and the fluxion liquid was collected. A total of 0.02 M PBS was used to balance the column again and was eluted with 0.1 M Glycine-HCL (pH = 2.5). The fluxion was collected and 1 M Tris-HCL (pH = 9) was added to it instantly. The samples above were dialyzed at 0.02 M PBS at 4 °C overnight and stored at −20 °C until use.

### 4.8. Preparation of SARS-CoV-2 Specific F(ab’)_2_

The immunoaffinity chromatography column was prepared in the early stage and combined with the total F(ab’)_2_ overnight at 4 °C. The next day, the penetrating fluid and the washing fluid were collected, respectively, and the sample was passed through 2–3 times after the components were collected.

Finally, the immunoaffinity chromatography column was eluted with 0.1 M of Glycine-HCl (pH = 2.7) and 1 M Tris-Hcl (pH = 10) was added to the eluent to prevent antibody inactivation. The samples above were dialyzed at 0.02 M PBS at 4 °C overnight and stored at −20 °C until use.

### 4.9. Comparison of the Affinity and Activity for Specific F(ab’)_2_ and Total F(ab’)_2_

ELISA and pseudovirus neutralization assay were used to detect the affinity and activity of the specific F(ab’)_2_ and the total F(ab’)_2_. For ELISA, briefly, SARS-CoV-2 RBD was antigen-coated with a 96-well plate and kept at 4 °C overnight. The board was washed with PBST five times, 5 min each time. The 96-well plate was sealed with 2% BSA for 2 h at 37 °C and washed with PBST. The special F(ab’)_2_, total F(ab’)_2_, and the control were diluted with a blocking solution in proportion and successively added to 96-well plates with 100 μL per well. Meanwhile, a negative control was set and incubated for 2 h at 37 °C. To detect the samples, 100 μL of antihuman IgG (Fab specific)-peroxidase (1:5000 dilution) was added to each well and incubated for 1 h at 37 °C and washed with PBST. TMB (100 μL/well) was then added. When the blue color was sufficiently visible, 50 μL of a 1.0 M HCl solution was added to each well to stop the enzymatic reaction. Finally, the absorbance was read at 450 nm.

For the pseudovirus neutralization assay, 293T cells were laid in a 6-well plate (0.7 × 10^6^ cells/mL) overnight until the cells covered about 80% of the plate. Then 3 mg pNL-4.3-R.E., 3 mg PcDNA3.1-RBD, and 12 μL P3000 with 8 μL Lip3000 were co-incubated for 20 min at room temperature. Finally, the mixtures were added to the plate for 48 h and the pseudovirus was collected 48 h later. Then the specific F(ab’)_2_, the total F(ab’)_2_, and the negative control in the same initial concentration were added to 96 wells. In the meantime, 293T-hACE2 (4 × 10^5^ cells/mL) and the pseudovirus were added to each well and the cell control and virus control were set at the same condition at 37 °C, 5% CO_2_, and 48 h. In the end, 100 μL luciferase solution containing lysate was added, and the absorbance was displayed by a microplate reader.

### 4.10. The Preparation of PEGylation F(ab’)_2_

A total of 10 mg TCEP was dissolved in a 1 mL buffer solution (20 mM sodium phosphate buffer, 50 mM NaCl, 10 mM EDTA, pH = 7.2), then 700 μL 10 mM TCEP was added to 2 mL specific F(ab’)_2_ (1 mg/mL) to react for 90 min at room temperature without shaking. In order to remove the TCEP completely when the reaction was complete, the process was repeated 4 times at 3500 rpm/min, 4 °C, 15 min each time and the buffer was exchanged with a 10 KDa ultrafiltration tube. Then the PEG was dissolved in a buffer solution to make the final concentration of 10 mg/mL, and the molar ratio of Fab:PEG was 1:1, 1:2, 1:4, 4 °C, 12 h, and no shaking.

### 4.11. Purification of PEGylated F(ab’)_2_

The prepared samples were purified by SP HP column. Firstly, the buffer of the PEGylated F(ab’)_2_ was replaced by an equilibration buffer (25 mM Tris-Hcl, pH = 8). The column was balanced with three column volumes of equilibration buffer and the load filtered samples to the column. Then five column volumes were used to wash the uncombined protein. Finally, the PEGylated F(ab’)_2_ fragment was eluted by applying a linear gradient of elution buffer (0–100% elution buffer in NaCl). The components of the eluent were collected for dialysis and concentration, stored at 4 °C until use, and identified in the end by SDS-PAGE.

### 4.12. Comparison of the Affinity and Activity for Specific F(ab’)_2_ and PEGylation F(ab’)_2_

The affinity was detected by ELISA. The special F(ab’)_2_, total F(ab’)_2_, and PEGylation F(ab’)_2_ were diluted to the same initial concentration and added to the 96-well plate. The other steps were the same as 2.8.1.

Briefly, the special F(ab’)_2_, total F(ab’)_2_, and PEGylated F(ab’)_2_ were diluted to the same initial concentration. In the meantime, the 293T-hACE2 (4 × 10^5^ cells/mL) and the pseudovirus were added to each well with a negative control, a cell control, and a virus control. The other steps were the same as 2.8.2.

### 4.13. Pharmacokinetics

Nine eight-week-old male SD (Sprague-Dawley) rats (240~260 g) were randomly allocated into three groups of five rats in each group. The three groups were the specific F(ab’)_2_, Fab-PEG-Fab, and Fab-PEG, respectively. The rats were subjected to a single intraperitoneal injection with the samples at 1 mg/kg body weight. The blood samples were collected from the rats at 30 min, 1, 2, 6, 12, 24, 36, 48, 72, and 96 h after injection. The concentrations of F(ab’)_2_ in the sera were determined by ELISA using an HRP-conjugated anti-F(ab’)_2_ polyclonal antibody and a known concentration of F(ab’)_2_ was taken as the standard. The T_1/2_ and AUC were measured by GraphPad Prime 8.

## 5. Conclusions

In summary, the preparation of PEGylated equine anti-SARS-CoV-2 specific F(ab’)_2_ not only retained most of the activity but also produced a longer half-life. Although equine F(ab’)_2_ fragments were heterologous proteins to the immune system, specific F(ab’)_2_ could minimize these effects. Additionally, PEGylated F(ab’)_2_ can prevent the reduced efficacy associated with high cost. It is these advantages that make it a candidate for the treatment of COVID-19.

## Figures and Tables

**Figure 1 ijms-24-03387-f001:**
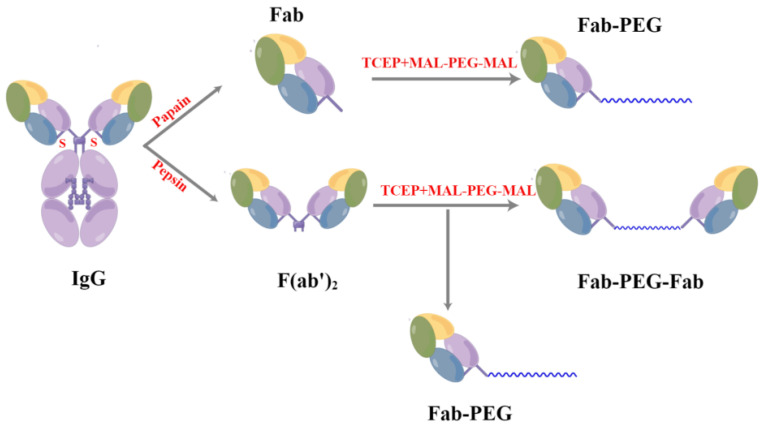
Structural representation of IgG, F(ab’)_2_, Fab-PEG, and Fab-PEG-Fab and the preparation process of PEGylated F(ab’)_2_. On the one hand, IgG was digested by pepsin to obtain F(ab’)_2_. F(ab’)_2_ was restored to Fab’ by TCEP and bound to 10 KDa MAL-PEG-MAL immediately. Fab-PEG and Fab-PEG-Fab were obtained. On the other hand, IgG was digested by papain to obtain Fab, and Fab bonded to 10 KDa MAL-PEG-MAL to obtain Fab-PEG.

**Figure 2 ijms-24-03387-f002:**
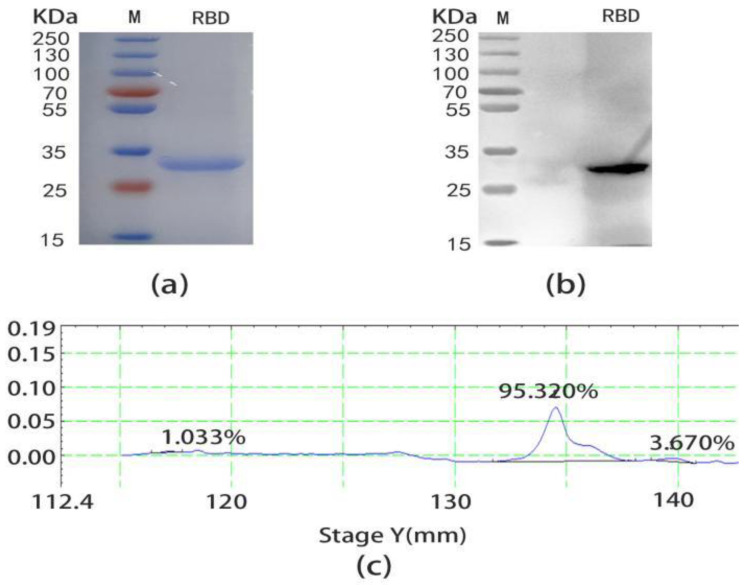
Expression and purification of SARS-CoV-2 RBD. (**a**) The SDS-PAGE of SARS-CoV-2 RBD induced at 37 °C for 5 h and purified by Ni-NTA. After mixing 20 μL SARS-CoV-2 RBD (10 mg) and 5 μL 5× loading buffer at 80 V for 20 min and 130 V for 40 min, decolorization took place after dyeing for 30 min. (**b**) The WB of SARS-CoV-2 RBD. SARS-CoV-2 RBD (10 mg) was transferred to a nitrocellulose membrane after SDS-PAGE, a rabbit anti-S1 polyclonal antibody was incubated at room temperature 2 h, and an HRP sheep anti-rabbit polyclonal antibody was diluted 1:10,000 at room temperature for 2 h. (**c**) Thin layer scanning of SARS-CoV-2 RBD. The purity of SARS-CoV-2 RBD was 95.320% and there were two impurities, 1.033% and 3.670%, respectively.

**Figure 3 ijms-24-03387-f003:**
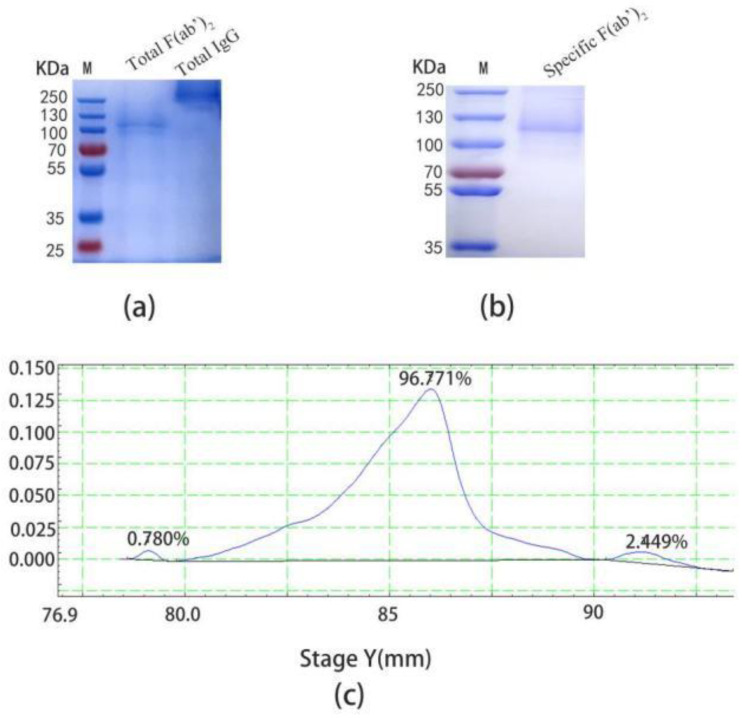
Preparation and purification of specific F(ab’)_2_. (**a**) Non-reduced SDS-PAGE of the total IgG (10 mg) and the total F(ab’)_2_ (10 mg). (**b**) Non-reduced SDS-PAGE of the specific F(ab’)_2_ (10 mg). (**c**) The purity of the specific F(ab’)_2_ by thin slice scanning, which was 96.771%. The other two impurities were 0.780% and 2.449%, respectively.

**Figure 4 ijms-24-03387-f004:**
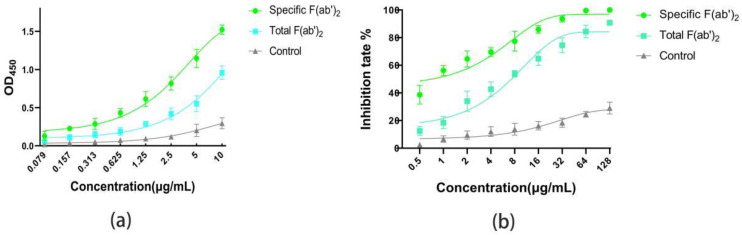
Comparison of the affinity and activity for specific F(ab’)_2_ and total F(ab’)_2_. (**a**) The absorbance at OD_450_ of the specific F(ab’)_2_ and the total F(ab’)_2_ by ELISA. (**b**) The inhibition rate of specific F(ab’)_2_ and total F(ab’)_2_ to the pseudovirus.

**Figure 5 ijms-24-03387-f005:**
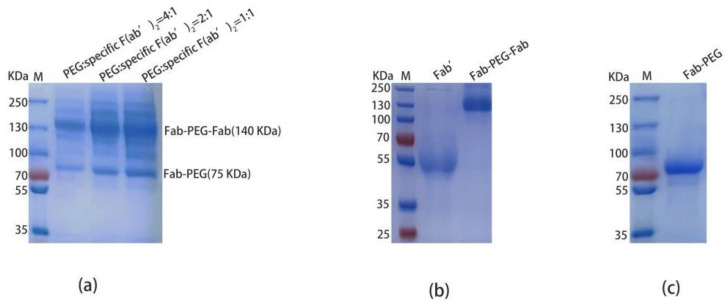
Non-reduced SDS-PAGE analysis of preparation and purification of PEGylated F(ab’)_2_. (**a**) The conditions of the specific F(ab’)_2_ (10 mg) binding with the MAL-PEG-MAL. Protein standards (line M); MAL-PEG-MAL: specific F(ab’)_2_ = 4:1 (lane 1); MAL-PEG-MAL: specific F(ab’)_2_ = 2:1 (lane 2); MAL-PEG-MAL: specific F(ab’)_2_ = 1:1 (lane 3). (**b**) The purification of the PEGylated F(ab’)_2_. Protein standards (line M); specific F(ab’)_2_ reduced by TCEP (lane 1); Fab-PEG-Fab (10 mg) was purified by ion chromatography (lane 2). (**c**) Protein standards (line M); Fab-PEG (10 mg) was purified by ion chromatography (lane 1).

**Figure 6 ijms-24-03387-f006:**
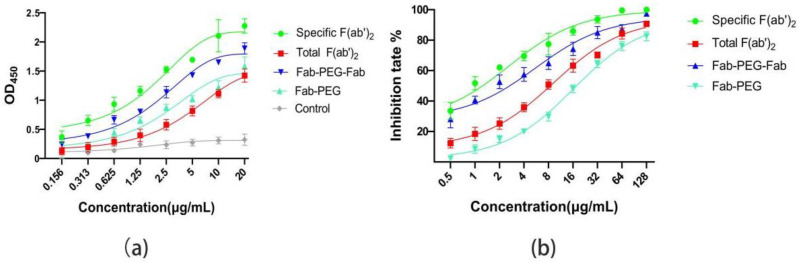
Comparison of the affinity and activity of specific F(ab’)_2_, PEGylated F(ab’)_2_, and total F(ab’)_2_. (**a**) The affinity of specific F(ab’)_2_, Fab-PEG-Fab, Fab-PEG, and total F(ab’)_2_ measured by ELISA. (**b**) The activity of specific F(ab’)_2_, Fab-PEG-Fab, Fab-PEG, and total F(ab’)_2_ measured by pseudovirus neutralization assay.

**Figure 7 ijms-24-03387-f007:**
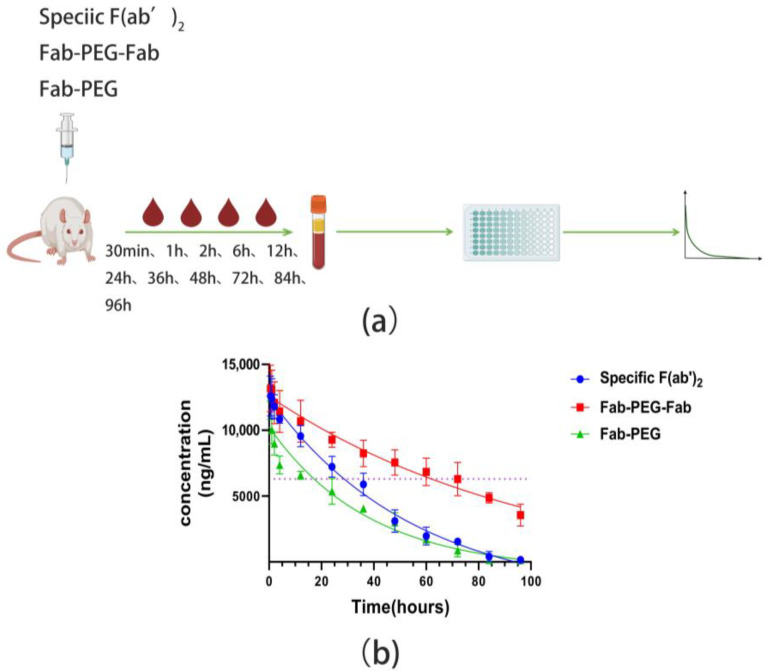
The detection of the half-life of the specific F(ab’)_2_, Fab-PEG-Fab, and PEG-Fab. (**a**) The scheme of collecting blood and testing the half-life. (**b**) Plot of plasma concentrations of specific F(ab’)_2_, Fab-PEG-Fab, and PEG-Fab obtained following single intravenous administration in SD rats at 1 mg/kg. The dotted line represents half of the highest concentration of the specific F(ab’)_2_.

**Table 1 ijms-24-03387-t001:** The PEGylated drugs on the market in recent years.

Product	Manufacturer	Drug	Year
Adagen	Enzon	Adenosine Deaminase (ADA)	1990
Oncasper	Enzon	Asparagine Synthase (ASNS)	1994
Doxil	Schering	Liposomes	1995
PEG-intron	Schering	IFN-α-2B	2000
PEGASYS	Roche	IFN-α-2α	2001
Oncasper	Enzon	Asparagine Synthase (ASNS)	1994
Doxil	Schering	Liposomes	1995
PEG-intron	Schering	IFN-α-2B	2000
PEGASYS	Roche	IFN-α-2α	2001
Neusalta	Amgen	G-CSF	2002
Somavert	Pfizer	Growth Hormone Antagonist (GHA)	2003
Mircera	Roche	Erythropoietin (EPO)	2007
Sylatron	Merck	IFN-α-2B	2011
Plegridy	Biogen	IFN-β-1α	2014
Adynovate	Baxalta	antihemophiliac globulin (AHG)	2015
Rebinyn	Novo Nordisk	Coagulation Factor IX	2017
Jivi	Bayer Healthcare	antihemophiliac globulin (AHG)	2018
Fulphila	Mylan GmbH	G-CSF	2018
Udenyca	Coherus Bioscience	G-CSF	2018
Ziextenzo	Sandoz	G-CSF	2019

**Table 2 ijms-24-03387-t002:** Recovery of specific F(ab’)_2_ and total F(ab’)_2_.

Sample	Concentration (mg/mL)	Recover Rates (%)
Total IgG	8.58	100
Total F(ab’)_2_	6.77	78.9
Special F(ab’)_2_	1.05	4.89

**Table 3 ijms-24-03387-t003:** Table of the main pharmacokinetic parameters calculated after intraperitoneal injection of specific F(ab’)_2_, Fab-PEG-Fab, and Fab-PEG.

Product	Dose (mg/kg)	Half-Life (h)
Specific F(ab’)_2_	1.0	38.32
Fab-PEG-Fab	1.0	71.41
Fab-PEG	1.0	26.73

## Data Availability

Not applicable.

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
