# Peer review of "PEGylation Prolongs the Half-Life of Equine Anti-SARS-CoV-2 Specific F(ab’)2"

_ijms, 2023, doi:10.3390/ijms24043387_

Round 1

Reviewer 1 Report

In this manuscript, Xu et al., developed the PEG-conjugated Fab. They also tested its anti-SARS-CoV-2 activity and half-life and showed that it had potent antiviral activity with long half-life. This new component might become a nice tool for the SARS-CoV-2 treatment. I have some comments for this manuscript.

Major

1.     The authors showed all figures with low resolution and it is hard/impossible to see the words in the graph and table. Please fix the resolution to high-resolution. In addition, the authors should use larger font.

2.     Figure 1 and Figure 7: In the figure the authors described “PEG-Fab-PEG”. However, the figure legend said “Fab-PEG-Fab”. Which is correct?

3.     Figure 1: the bottom figure looks Fab-PEG. However, the authors described PEG-Fab-PEG. Is it correct?

4.     Figure1: there are no explanation about Papain-digested pathway. The authours should at least explain it in the result.

5.     Figure 2a: The protein band from Total IgG looks cut off (>250kDa). The authors should show uncut band.

6.     Figure 2, 3, 5: For the SDS-PAGE and WB, how much (mg) of proteins were used? Please clarify in the Figure legend.

7.     Figure 2c and Figure 3d: What the table in the bottom means? Please put the high-resolution picture and explain it in the legend.

8.     Line 288-295: these sentences are about Figure 4. The authors cite Figure 5. Please fix them. 

9.     Figure 6a: the authors used both Fab-PEG-Fab and PEG-Fab-PEG. Why they used both? Or PEG-Fab-PEG is typo?

10.  Figure 6b: Did the authors test whether this antiviral activity was SARS-CoV-2 specific?

11.  Please add the reference: Line 65-68. Line73-75. Line81-82.

12.   

Minor

1.     Figure 1: Please delete the space in F(ab’  )2

2.     Figure 7b: “PEG-Fab” should “Fab-PEG”.

3.     Line 261: Figure 2d should be Figure 3d.

4.     Line 261: Figure 5a should be Figure 6a.

5.     Line 299: “Pseudovirusin” is typo.

6.     Line 363, 370: “SARS-COV-2” should be “SARS-CoV-2”.

7.     Line 319: the font size is bigger than others. Please fix it.

Author Response

Reviewer 1

Comments and Suggestions for Authors

In this manuscript, Xu et al., developed the PEG-conjugated Fab. They also tested its anti-SARS-CoV-2 activity and half-life and showed that it had potent antiviral activity with long half-life. This new component might become a nice tool for the SARS-CoV-2 treatment. I have some comments for this manuscript.

Major

  1. The authors showed all figures with low resolution and it is hard/impossible to see the words in the graph and table. Please fix the resolution to high-resolution. In addition, the authors should use larger font.

The all figures have been changed.

  1. Figure 1 and Figure 7: In the figure the authors described “PEG-Fab-PEG”. However, the figure legend said “Fab-PEG-Fab”. Which is correct?

PEG-Fab-PEG in Figure 7 have changed to Fab-PEG-Fab.

  1. Figure 1: the bottom figure looks Fab-PEG. However, the authors described PEG-Fab-PEG. Is it correct

PEG-Fab-PEG have changed to Fab-PEG in Figure 1.

  1. Figure1: there are no explanation about Papain-digested pathway. The authours should at least explain it in the result.

 Papain-digested pathway have full in.

  1. Figure 2a: The protein band from Total IgG looks cut off (>250kDa). The authors should show uncut band.

Figure 2a have changed.

  1. Figure 2, 3, 5: For the SDS-PAGE and WB, how much (mg) of proteins were used? Please clarify in the Figure legend.

The all proteins for SDS-PAGE and WB were 10mg.

  1. Figure 2c and Figure 3d: What the table in the bottom means? Please put the high-resolution picture and explain it in the legend.

Figure 2c and Figure 3d were the purity of SARS-CoV-2 RBD and specific F(ab’)2.In addition,the high-resolution picture have modified and explained in the figure lengend.

  1. Line 288-295: these sentences are about Figure 4. The authors cite Figure 5. Please fix them. 

I have changed it.

  1. Figure 6a: the authors used both Fab-PEG-Fab and PEG-Fab-PEG. Why they used both? Or PEG-Fab-PEG is typo?

PEG-Fab-PEG is wrong,PEG-Fab-PEG have changed to Fab-PEG.

  1. Figure 6b: Did the authors test whether this antiviral activity was SARS-CoV-2 specific?

Yes,I did.I adopted the principle of specific binding of antigen and antibody, and used SARS-CoV-2 RBD as antigen to detect the specificity of specific F(ab')2.

  1. Please add the reference: Line 65-68. Line73-75. Line81-82.

I had added three references to the manuscript.

  1.  

Minor

  1. Figure 1: Please delete the space in F(ab’)2

I have already deleted the space in F(ab’)2.

  1. Figure 7b: “PEG-Fab” should “Fab-PEG”.

I have changed PEG-Fab to Fab-PEG.

  1. Line 261: Figure 2d should be Figure 3d.

I have changed Figure 2d to Figure 3d.

  1. Line 261: Figure 5a should be Figure 6a.

I have changed Figure 5a to Figure 6a.

  1. Line 299: “Pseudovirusin” is typo.

I have changed Pseudovirusin to Pseudovirus.

  1. Line 363, 370: “SARS-COV-2” should be “SARS-CoV-2”.

I have changed SARS-COV-2 to SARS-CoV-2.

  1. Line 319: the font size is bigger than others. Please fix it.

I have changed it.

Reviewer 2 Report

This study by Xu et al was done to assess whether PEGylated F(ab’)2 of equine antibodies obtained through immunization, would demonstrate efficient antibody binding and neutralization activities whilst exhibiting potential for improved (longer) half-lives as compared to non-PEGylated versions. The work was also done to address whether such therapies could be added to the armour of antivirals currently available for targeting SARS-CoV-2. Such approaches are important as they provide additional information pertaining to the utility of small molecule and in particular redesigned antibody therapies.

This is a good study however a few r suggestions below for the authors consideration.

Major

-       Descriptions of assays, in particular neutralization assay, seems very vague and difficult to follow. Were 293ThACE2 cells plated overnight prior to neutralization as is commonly done or does the group plate cells and conduct neutralization at the same time. Furthermore, were the F(ab’)2 and pseudovirions incubated prior to addition to 293ThACE2. Please clarify this as the set up can and does impact results/outcomes.

-       Not entirely sure which figure the authors refer to at line 305/306. Authors use the term ‘crave’ and indicate Figure 5 which is a western blot. Please clarify if the ‘crave’ referred to here is indeed the gel image or do the authors mean the ELISA data (curves that are in Figure 6a). While not a major comment, it is imperative that the authors clarify this.

-       It might be important for the authors to also assess neutralization of VOCs as well the original Wuhan strain to perhaps highlight its ability to provide broad protection.

-       Whilst perhaps minor, it would also be important for the authors to perhaps perform authentic virus neutralization assays to strengthen their results

Minor

-       The authors state that the use of F(ab’)2 favourably reduced ADCC and CDC, however, the benefits of ADCC in controlling COVID has been documented by numerous studies including its contribution to protection even with convalescent plasma (Begin et al 2022). Perhaps reword to indicate some Fc functions can indeed be harmful with CDC as the example.

-       Line 16: grammar, remove ‘of’ after small sized

-       Line 17: Maximize as opposed to maximum

-       Line 56: space to be added between reactioncaused

-       Line 73: rewrite as …’greatly reduces or even inactivates the activity’ to prevent a hanging sentence.

-       Line 148: change eluented to ‘used to elute’

-       Line 172: change eluention to eluent/eluate

-       Line 177: typo of ‘tatal’. Change to total

-       Line 191: typo of ‘nagative’. Change to negative

I believe this study demonstrates that PEGylated F(ab’)2 has tremendous potential. It is also encouraging that the authors are able to show this strategy improving significantly the half-life of F(ab’)2 and thus likely promising as a cost-effective approach.  Addressing the concerns above will further improve this work which is indeed quite promising.

Author Response

Reviewer 2

Comments and Suggestions for Authors

This study by Xu et al was done to assess whether PEGylated F(ab’)of equine antibodies obtained through immunization, would demonstrate efficient antibody binding and neutralization activities whilst exhibiting potential for improved (longer) half-lives as compared to non-PEGylated versions. The work was also done to address whether such therapies could be added to the armour of antivirals currently available for targeting SARS-CoV-2. Such approaches are important as they provide additional information pertaining to the utility of small molecule and in particular redesigned antibody therapies.

This is a good study however a few r suggestions below for the authors consideration.

Major

-       Descriptions of assays, in particular neutralization assay, seems very vague and difficult to follow. Were 293ThACE2 cells plated overnight prior to neutralization as is commonly done or does the group plate cells and conduct neutralization at the same time. Furthermore, were the F(ab’)and pseudovirions incubated prior to addition to 293ThACE2. Please clarify this as the set up can and does impact results/outcomes.

I have changed it again.

-       Not entirely sure which figure the authors refer to at line 305/306. Authors use the term ‘crave’ and indicate Figure 5 which is a western blot. Please clarify if the ‘crave’ referred to here is indeed the gel image or do the authors mean the ELISA data (curves that are in Figure 6a). While not a major comment, it is imperative that the authors clarify this.

 Sorry,the crave that I describe is Figure 6a.I wrote Figure 6a as Figure 5 wrongly.

-       It might be important for the authors to also assess neutralization of VOCs as well the original Wuhan strain to perhaps highlight its ability to provide broad protection.

 Yes,the original Wuhan strain as is important for my study. VOCs isn’t assessed because we didn't have what it took.

-       Whilst perhaps minor, it would also be important for the authors to perhaps perform authentic virus neutralization assays to strengthen their results

Minor

-       The authors state that the use of F(ab’)2 favourably reduced ADCC and CDC, however, the benefits of ADCC in controlling COVID has been documented by numerous studies including its contribution to protection even with convalescent plasma (Begin et al 2022). Perhaps reword to indicate some Fc functions can indeed be harmful with CDC as the example.

-       Line 16: grammar, remove ‘of’ after small sized

 I have changed it.

-       Line 17: Maximize as opposed to maximum

 I have changed it.

-       Line 56: space to be added between reactioncaused

 I have changed it.

-       Line 73: rewrite as …’greatly reduces or even inactivates the activity’ to prevent a hanging sentence.

 I have changed it.

-       Line 148: change eluented to ‘used to elute’

 I have changed it.

-       Line 172: change eluention to eluent/eluate

 I have changed it.

-       Line 177: typo of ‘tatal’. Change to total

 I have changed it.

-       Line 191: typo of ‘nagative’. Change to negative

  I have changed it.

I believe this study demonstrates that PEGylated F(ab’)has tremendous potential. It is also encouraging that the authors are able to show this strategy improving significantly the half-life of F(ab’)2 and thus likely promising as a cost-effective approach.  Addressing the concerns above will further improve this work which is indeed quite promising.

Reviewer 3 Report

In this manuscript, Xu et al investigated the impact of PEGylation on the half-life of anti-SARS-CoV-2 F(ab’)2. The study is well conducted and the manuscript is overall pleasant to read.

I have specific points to address:

Unless I am mistaken, the authors did not explain how Equine anti-SARS-CoV-2 serum was obtained. It would be interesting to specify it, at least in the Materials & Methods.

Line 261: change Figure 2d by Figure 3d

Concerning the Pseudovirus neutralization assays, the authors do not specify the nature of the RBD domain used (from which variant?). It would seem relevant or even critical to perform this type of test with a RBD domain of an Omicron subvariant if this has not been done.

Typos :

Section 2.7. Purification of total F (ab ') 2: change Brifely by Briefly

Line 217: change Brifly by briefly

Line 270: change assayt by assays

Line 290: change attched by attached

Author Response

Dear Editor and Reviewer

Thank you for your comments concerning ourmanuscript entitled “PEGylation prolongs the half-life of equine anti-SARS-CoV-2 specific F(ab’)2. Those comments are all valuable and very helpful for revising andimproving our paper, as well as the important guiding significance toour researches.  We have studied comments carefully and have madecorrection which we hope meet with approval.

Comments and Suggestions for Authors

In this manuscript, Xu et al investigated the impact of PEGylation on the half-life of anti-SARS-CoV-2 F(ab’)2. The study is well conducted and the manuscript is overall pleasant to read.

I have specific points to address:

Unless I am mistaken, the authors did not explain how Equine anti-SARS-CoV-2 serum was obtained. It would be interesting to specify it, at least in the Materials & Methods.

Thanks for your evaluation. The equine anti-SARS-CoV-2 serum was obtained by immunogen immunized the horse.Immunogens are bacterial-like particles with RBD as its component.Horses were immunized at a dose of 3mg each time, and blood was collected after five consecutive immunizations every other week following the first immunization.Unfortunately, the relevant data and evaluation have been published in another article, so we can not give too much detail.

Line 261: change Figure 2d by Figure 3d

I have changed it.

Concerning the Pseudovirus neutralization assays, the authors do not specify the nature of the RBD domain used (from which variant?). It would seem relevant or even critical to perform this type of test with a RBD domain of an Omicron subvariant if this has not been done.
Thanks for your evaluation.At the Pseudovirus neutralization assays, the RBD was from D614.Relevant data are supplemented in the manuscript

Typos :

Section 2.7. Purification of total F (ab ') 2: change Brifely by Briefly

I have changed it.

Line 217: change Brifly by briefly

I have changed it.

Line 270: change assayt by assays

I have changed it.

Line 290: change attched by attached

I have changed it.

Round 2

Reviewer 1 Report

I think the authors answered all my questions.

Reviewer 3 Report

All concerns have been clarified by the authors